# The Association between Red Meat Consumption and Advanced Colorectal Adenomas in a Population Undergoing a Screening-Related Colonoscopy in Alberta, Canada

**DOI:** 10.3390/cancers16030495

**Published:** 2024-01-24

**Authors:** Eliya Farah, John M. Hutchinson, Yibing Ruan, Dylan E. O’Sullivan, Robert J. Hilsden, Darren R. Brenner

**Affiliations:** 1Department of Oncology, Cumming School of Medicine, University of Calgary, Calgary, AB T2N 4N1, Canada; eliya.farah1@ucalgary.ca (E.F.); john.hutchinson@ucalgary.ca (J.M.H.); dylan.osullivan@ucalgary.ca (D.E.O.); 2Department of Community Health Sciences, Cumming School of Medicine, University of Calgary, Calgary, AB T2N 4N1, Canada; yibing.ruan@albertahealthservices.ca (Y.R.); rhilsden@ucalgary.ca (R.J.H.); 3Department of Cancer Epidemiology and Prevention Research, Cancer Care Alberta, Calgary, AB T2N 4N1, Canada; 4Department of Medicine, Cumming School of Medicine, University of Calgary, Calgary, AB T2N 4N1, Canada; 5Forzani & MacPhail Colon Cancer Screening Centre, Alberta Health Services, Calgary, AB T2N 4Z6, Canada

**Keywords:** processed red meat, unprocessed red meat, risk factor, advanced colorectal adenomas, colorectal cancer

## Abstract

**Simple Summary:**

In this study, we examined whether different types of red meat, processed and unprocessed, contribute to the development of advanced colorectal adenomas, which can be precursors to cancer. Through examining the diets of 1083 individuals at their initial colonoscopy in Calgary, we assessed the impact of meat intake on the presence of these early cancer indicators. We observed that the total amount of red and processed meat, but not unprocessed meat, was associated with the presence of advanced colorectal adenomas at the time of a screening colonoscopy. Our observations provide preliminary evidence that may contribute to a nuanced understanding of diet’s role in colorectal health.

**Abstract:**

The association between red meat consumption and colorectal cancer has been rigorously examined. However, a more comprehensive understanding of how the intake of unprocessed red meat contributes to the development of early precancerous colorectal lesions, such as advanced colorectal adenomas (ACRAs), requires further investigation. We examined the associations between different types of red meat intake and ACRAs in a sample population of 1083 individuals aged ≥ 50 years undergoing an initial screening colonoscopy in Calgary, Alberta, Canada. Associations between grams per day of total, processed, and unprocessed red meat from diet history questionnaires and ACRAs were evaluated with multivariable logistic regression models. We also applied cubic spline models fitted with three knots (10th, 50th, and 90th percentiles) to identify potential nonlinear associations. We did not observe a meaningful association between unprocessed red meat intake and the presence of ACRAs. In contrast, for every 10 g/d increase in total and processed meat intake, we observed an increase in the odds of ACRAs at the screening colonoscopy (adjusted odds ratio (OR) = 1.05, 95% [CI = 1.01–1.09], *p* = 0.04) and (adjusted OR = 1.11, 95% [CI = 1.02–1.20], *p* = 0.02), respectively. This study highlights the importance of differentiating between types of red meat consumption in the context of dietary risks associated with ACRAs.

## 1. Introduction

Colorectal cancer (CRC) is the fourth-most diagnosed cancer in Canada, accounting for approximately 10% of all cancer cases. It was estimated that 24,100 Canadians were diagnosed with CRC and 9400 died of the disease during 2023 [1]. Among the various modifiable risk factors associated with CRC, diet, particularly the consumption of red meat, has garnered considerable interest due to its potential role in increasing CRC risk [2]. This focus on diet and its impact on CRC risk intersects with the classification of meats based on their fat, cholesterol, and iron content. Meats are commonly classified into two categories based on their fat, cholesterol, and iron contents: red meats (such as beef, pork, and lamb) and white meats (including chicken and turkey). Additionally, meat can be further divided into processed meat, which is preserved using high levels of salt and/or chemical additives, and unprocessed meat, which is consumed without such preservatives [3,4].

The association between red meat consumption and CRC has been investigated by numerous studies [5,6,7,8,9,10,11]. The World Cancer Research Fund and the American Institute for Cancer Research established a strong association between a high consumption of red and processed meat and an increased risk of CRC [12]. Similarly, the International Agency for Research on Cancer has classified processed red meat as a human carcinogen, while suggesting that red meat could possibly have similar effects [13,14]. While the general consensus indicates processed meats as carcinogenic, with red meats potentially presenting similar risks, there is a lack of consensus on the risks associated with unprocessed red meat consumption [13,14,15,16]. For instance, the Burden of Proof study from the Global Burden of Disease Team observed weak evidence associating unprocessed red meat consumption to an increased risk of CRC [16]. Therefore, further research is needed to clarify the role of meat processing in the risk of CRC development.

Inconsistencies in the existing literature may be attributed to the inadequate distinction between processed and unprocessed red meat, because the health implications of each type can vary significantly [17,18]. Unprocessed red meat can serve as a lean source of dietary protein and generally contains fewer calories and less fat and cholesterol per serving compared to processed meat [19]. Moreover, processed meats typically contain a high concentration of salts and nitrates, among other preservatives, which are substances that have been linked to potential adverse health outcomes [20]. As such, it may not be justified to group processed and unprocessed meats together without evidence supporting such a classification [4]. In addition, emerging research highlights the importance of considering the type and preparation of red meat, because these factors may play a more significant role in determining the risk of CRC [21]. While much of the existing literature has investigated the relationship between red meat intake and CRC, it is equally important to understand the relationship with its precursor, advanced colorectal adenomas (ACRAs) [8,9,22]. This represents an important area that necessitates further investigation to enhance our understanding of the disease’s early development stages.

This study aims to further evaluate this relationship by examining the association between different types of red meat intake (total, processed, and unprocessed) and early precancerous colorectal lesions among a sample of individuals undergoing a screening colonoscopy in Calgary, Alberta, Canada.

## 2. Materials and Methods

### 2.1. Study Population

We examined data from 1083 participants aged ≥ 50 years, who completed the Diet History Questionnaire (DHQ), and underwent a screening colonoscopy during 2008–2020 at the Forzani & MacPhail Colon Cancer Screening Centre (CCSC). The CCSC is an endoscopy center that provides screening-related colonoscopies in the Calgary region, Alberta, Canada. Individuals are referred to the CCSC by their primary healthcare provider for one or more of the following reasons: routine CRC screening among average-risk individuals, a personal or family history of adenomatous polyps or CRC, or a positive CRC test result from an alternative screening test.

We used the CCSC referral database to identify participants at average risk for CRC, defined as individuals with no first-degree relative (parent, sibling, or child) of CRC, and invited them to participate. We recruited all other participants during their initial medical assessment at the CCSC, which takes place a few weeks before their colonoscopy procedure. The cohort profile of the Forzani & MacPhail CCSC biorepository has been further describe in Hilsden et al. [23].

For our noncases group, these participants were required to be above 50 years of age, at an average risk of CRC (no family history of CRC), have undergone a complete colonoscopy with adequate bowel preparation, and importantly, they could not have a history of high-risk serrated polyps. In contrast, our case group consisted of individuals above 50 years of age who were diagnosed with an ACRA during their colonoscopy procedure. Ethics approval for data collection and recruitment was obtained from the University of Calgary Conjoint Health Research Ethics Board (CHREB) and was eventually transferred to the Health Research Ethics Board of Alberta Cancer Committee (ID HREBA.CC-16-0224).

### 2.2. Data Collection

In the context of our paper, unprocessed red meat refers to unprocessed muscle meat sourced from mammals, including but not limited to beef, veal, pork, and lamb. This category encompasses these meats in various forms, such as minced or frozen. Processed meat, on the other hand, involves meat that has undergone modifications aimed at enhancing flavor or extending shelf life. These alterations may include processes like salting, curing, fermentation, smoking, or other preservation methods [4].

Diet, health, and lifestyle data were collected prior to the completion of the colonoscopy and result ascertainment. We collected data on red meat, sugar, fat, and protein dietary intake from the Diet History Questionnaire (DHQ I & II) [24,25] and used the NIH-developed software Diet*Calc (v 1.5.0) to compute daily intake of macronutrients such as protein, fat, and carbohydrates in grams, as well as calories in kilocalories [26]. We measured the level of physical activity using the International Physical Activity Questionnaire (IPAQ) [27,28]. This instrument comprehensively evaluates physical activity across various domains, encompassing leisure time, domestic tasks, gardening (yard) activities, work-related engagements, and transportation-related activity. It covers specific types of activities, including walking, moderate-intensity exercises, and vigorous intensity activities. To estimate the overall physical activity level, we collected separate data on the frequency (measured in days per week) and duration (time per day) for each specific activity type. We also collected information on health, comorbidities, alcohol consumption, and smoking status. Following the colonoscopy, the endoscopist documented the size, location, and shape of any identified polyp(s). Then, a trained research assistant reviewed the pathology report and assigned a histological category to each identified polyp.

Within the 165 questions included in the DHQ, 22 examined the consumption of red meat (i.e., How often did you eat hot dogs, wieners, or frankfurters? How often did you eat steak (beef)? Each time you ate roast beef or steak in sandwiches, how much did you usually eat? Each time you ate beef hamburgers or cheeseburgers from a fast food or other restaurant, how much did you usually eat?) [29]. The key exposure variables for this study were the daily intake of total, processed, and unprocessed red meat, measured in grams. Unprocessed red meat intake encompassed foods such as steaks, roasts, ground beef, beef mixtures, and spareribs, along with pork. Processed meat intake was estimated via summing the intake of red meats that have been processed using techniques such as salting, smoking, or curing. This category included foods like hot dogs, bacon, sausages, cheese hamburgers, deli-style meats, or baked ham [24,25].

Our final data set comprised 1083 individuals—after the exclusion of 2992 individuals who either did not complete the DHQ or had missing data on red meat intake or screening outcomes. To minimize the potential impact of outlying observations, we truncated total energy intake (kcal/d) values at both the 99th (574.9 kcal/d) and 1st (4396 kcal/d) percentiles. Furthermore, we truncated red meat intake, including total (283 g/d), processed (107 g/d), and unprocessed (218 g/d) only at the 99th percentile, while abstaining from truncation at the 1st percentile to account for individuals adhering to dietary lifestyles that exclude red meat consumption, such as vegans, vegetarians, and pescatarians. ACRAs were defined and classified as polyps (1) of a size of 1 cm or more as recorded by the endoscopist, (2) a villous structure revealed using histology, or (3) the presence of high-grade dysplasia [30].

### 2.3. Statistical Analysis

We employed multivariable logistic regression models using the glm R package (v4.2.1) with crude and adjusted odds ratios (ORs) with their 95% confidence intervals (Cis) to estimate the association between red meat consumption and CRA development.

We quantified red meat consumption using a standard unit of 10 g. The first model (M1) was unadjusted. The second model (M2) adjusted for sex (males/females) and age (continuous). The third model (M3) adjusted for M2 as well as body mass index (BMI) (<25, 25–30, 30 kg/m^2^), smoking status (never, former, current smoker), alcohol consumption (daily, former, occasional, never), ethnicity (white, other), diabetes (yes/no), and physical activity (inactive, low, moderate, and high activity). The fourth model (M4) adjusted for M3 as well as total energy intake (kcal/d), and protein and fat intake (g/d).

We evaluated potential interactions between the red meat variables, smoking, and BMI in our linear regression model using multiplicative interaction terms. However, none of these interactions achieved significance at the *p* < 0.1 level and therefore were not included in the manuscript.

To evaluate potential nonlinear associations in the relationship between CRA and red meat consumption, we employed a three-knot restricted cubic spline using the plotRCS (v0.1.4) R package for red meat intake. The cubic spline model provides a flexible method to illustrate the association and was fitted with three knots positioned at the 10th, 50th, and 90th percentiles, using the 10th percentile as the reference. This method allows for a potential change in the direction of the association at the knot points, thus highlighting any nonlinear relationship between the variables that might otherwise be missed in standard log-linear models.

## 3. Results

### 3.1. Study Population

Table 1 presents the characteristics of the study participants, including demographic information, health parameters, and lifestyle factors. Among the 1083 individuals included in the analysis, 215 (20%) had one or more ACRA detected. The average age at enrollment was 60, and 35% of participants were women, and 86% were white. The average daily consumption included 73 g of meat, broken down as 49 g unprocessed and 24 g processed (Figure 1). The daily energy consumption averaged 1765 kcal, with a fat intake of 66 g, protein intake of 72 g, and sugar intake of 94 g. In addition, 60% of participants reported daily alcohol use and 62% reported high physical activity. Also, 10%, 43%, and 47% of participants were current, former, and never smokers, respectively. Finally, 10% of participants reported a diabetes diagnosis and 70% reported a BMI of greater than or equal to 25 kg/m^2^ (Table 1).

### 3.2. Model Outcomes

Table 2 provides a summary of the logistic regression results for total, processed, and unprocessed red meat intake, all analyzed per 10 g increases. Across all models, we observed that for every 10 g increase in total red meat intake, the odds of developing CRA increased: Model 1 (crude OR = 1.03, 95% [CI = 1.01–1.06]), Model 2 (OR = 1.03, 95% [CI = 1.01–1.06]), Model 3 (OR = 1.02, 95% [CI = 0.99–1.05]), and Model 4 (OR = 1.05, 95% [CI = 1.01–1.09]) (Table 2). Similarly, unprocessed meat intake was associated with a modest increase in the odds of developing ACRAs across all models: Model 1 to 4 (OR ranging from 1.02 to 1.04). On the contrary, for every 10 g increase in processed meat intake, we observed a pronounced increase in the odds of developing CRA in all models: Model 1 (OR = 1.12, 95% [CI = 1.05–1.20]), Model 2 (OR = 1.11, 95% [CI = 1.04–1.18]), Model 3 (OR = 1.09, 95% [CI = 1.01–1.17]), and Model 4 (OR = 1.12, 95% [CI = 1.03–1.22]) (Table 2).

The results from our cubic spline analyses align with the findings from our logistic models. We observed no association between total meat intake and the development of ACRAs (*p* = 0.103), or in the nonlinear term (*p* = 0.879) (Figure 2A). Similarly, unprocessed meat intake showed no associations in both the overall (*p* = 0.280) and nonlinear model terms (*p* = 0.351) (Figure 2B). Processed meat intake showed a dose-response relation with the ACRAs (*p* = 0.011), which did not depart from linearity (*p* for nonlinearity: 0.169) (Figure 2C).

## 4. Discussion

In this study, we aimed to explore the association between different types of red meat intake—total, processed, and unprocessed—and the development of early precancerous colorectal lesions (advanced colorectal adenomas, ACRAs) among a sample of individuals undergoing a screening colonoscopy in Calgary, Alberta, Canada. Unlike total and processed red meat, we did not observe significant associations between ACRAs and unprocessed meat consumption, possibly due to this study’s limited sample size. These preliminary findings necessitate further research with larger sample sizes to better understand the impact of different types of meat consumption on colorectal carcinogenesis.

Our research findings align with an array of literature that has explored the relationship between red meat consumption and ACRA development [3,5,6,7,8,9,10,11,31]. Consistent with our findings, several studies have demonstrated an increased association between processed meat intake and colorectal adenomas [5,6,8,9,12,13]. Similarly, concerning unprocessed red meat, our results align with findings from the Burden of Proof study and other related research, which present weak evidence for an association with ACRA [5,7,10,16,32]. These findings may imply that the consumption of unprocessed, lean, red meats could potentially be part of a healthy diet conscious of cancer risk. While this study does not contradict the classification of red meat as a probable carcinogen, it highlights the need for further investigations to elucidate the circumstances under which red meat consumption might elevate ACRA and CRC risk, such as specific quantities, preparation methods, and accompanying dietary or lifestyle risk factors.

Emerging evidence has highlighted the importance of considering not only the type of red meat but also the method of preparation [33,34,35]. The manner in which meat is cooked, the temperature it reaches, and the duration for which it is cooked can affect the formation of potentially carcinogenic compounds [36]. In addition, processed meats often contain higher amounts of carcinogens, such as N-nitroso compounds (NOCs), heterocyclic amines (HCAs), and polycyclic aromatic hydrocarbons (PAHs), which are formed during processing or cooking at high temperatures [37]. These meats are often distinguished through high levels of added nitrates and nitrites, preservatives that can convert into carcinogenic nitrosamines within the body, setting them apart from unprocessed meats [38]. In addition, high-temperature cooking methods, such as grilling or barbecuing, have been shown to elevate HCA and PAH levels [37,39]. Further, Zheng et al., demonstrated that consumption of well-done processed meat, a result of extended cooking times, was associated with increased CRA risk, possibly due to elevated levels of HCAs [39]. These findings emphasize the necessity to consider meat preparation and cooking methods when assessing the impacts of red meat on CRA risk. Future studies should examine the risk of CRA and CRC associated with the consumption of unprocessed meats through accounting for preparation methods such as high-temperature cooking.

This study has several limitations. First, although we have adjusted for confounding variables in our analyses, the potential for residual confounding remains. Second, this study’s relatively small sample size might result in a large variability of effect estimates. Third, our self-reported dietary data could introduce reporting bias and measurement error, potentially leading to inaccurate results. Lastly, the cross-sectional nature of this study limits our ability to establish a temporal relationship between red meat consumption and ACRAs because our questionnaire did not collect data pertaining to past dietary habits. Despite these limitations, this study contributes valuable insights to the ongoing discussion on the relationship between red meat intake and ACRAs.

## 5. Conclusions

This study provides key insights into the relationship between red meat consumption and ACRAs. While not directly contradicting the classification of red meat as a probable carcinogen, our results underscore the importance of further investigations into the specific contexts in which red meat consumption may elevate ACRA and CRC risk, such as quantity, preparation methods, and associated dietary or lifestyle factors. As this discourse continues to evolve, it is important to acknowledge that the outright avoidance of unprocessed red meat may fail to consider the benefits offered by lean, unprocessed meats in a balanced diet. Therefore, this study contributes to the discussion surrounding the prevalent perception of red meat consumption and its potential adverse health implications. Through highlighting these complexities, we advocate for a more nuanced and comprehensive examination of the multifaceted role of red meat in health and cancer risk.

## Figures and Tables

**Figure 1 cancers-16-00495-f001:**
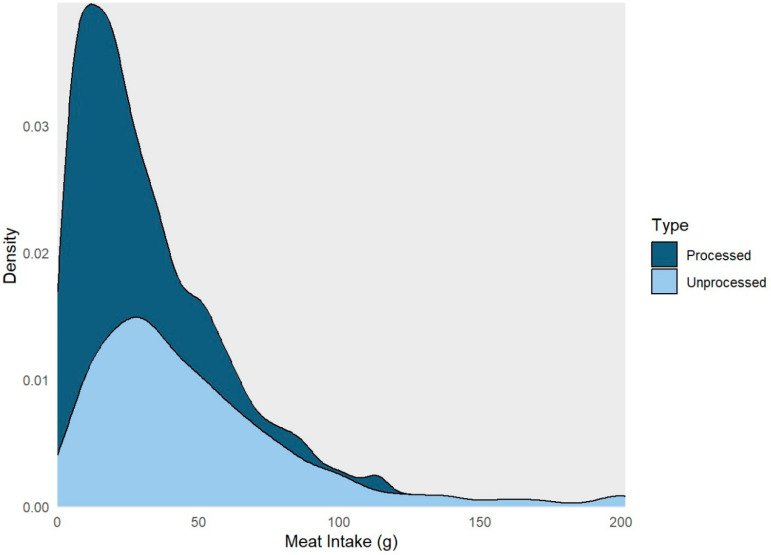
Density plot highlighting the distribution of processed and unprocessed red meat intake among the study participants, aged ≥ 50 years, who completed the Diet History Questionnaires, and underwent a screening colonoscopy during 2008–2016 at the Forzani & MacPhail Colon Cancer Screening Centre (*n* = 1083).

**Figure 2 cancers-16-00495-f002:**
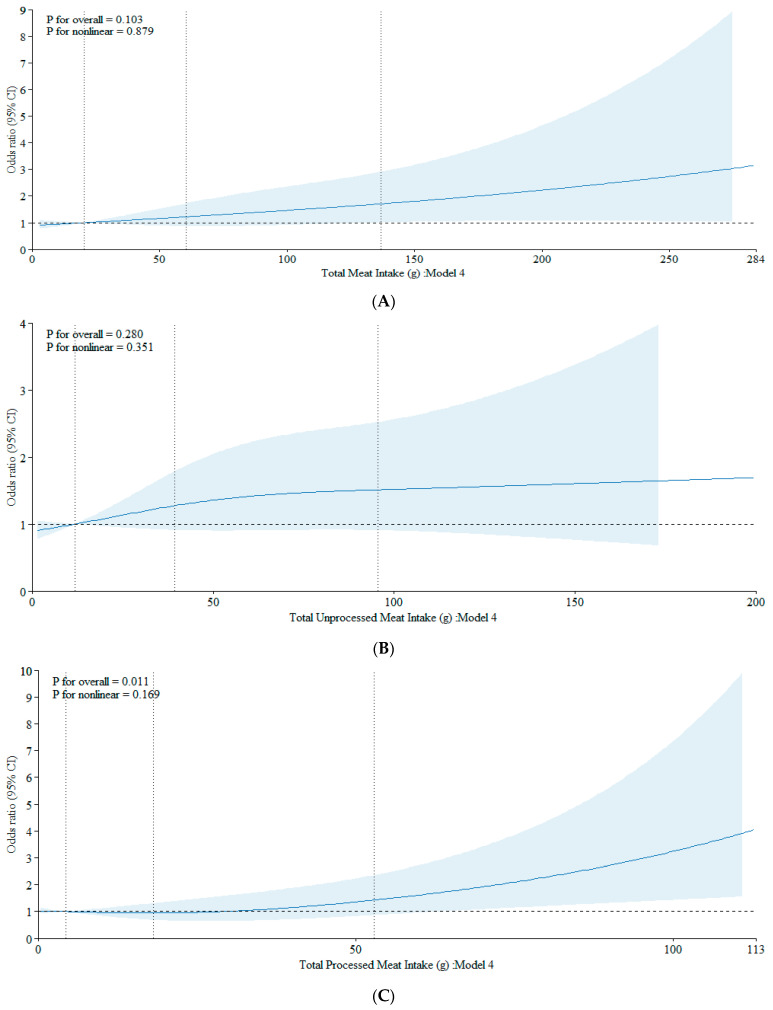
(**A**) Association between total meat intake and CRA using a restricted cubic spline regression model. The data were fitted with three knots at the 10th, 50th, and 90th percentiles (with the reference being the 10th percentile). The solid lines represent the ORs, while the shaded areas indicate the 95% CI. (**B**) Association between unprocessed meat intake and CRA using a restricted cubic spline regression model. The data were fitted with three knots at the 10th, 50th, and 90th percentiles (with the reference being the 10th percentile). The solid lines represent the ORs, while the shaded areas indicate the 95% CI. (**C**) Association between processed meat intake and CRA using a restricted cubic spline regression model. The data were fitted with three knots at the 10th, 50th, and 90th percentiles (with the reference being the 10th percentile). The solid lines represent the ORs, while the shaded areas indicate the 95% CI.

**Table 1 cancers-16-00495-t001:** Baseline characteristics of the study participants, aged ≥ 50 years, who completed the Diet History Questionnaires, and underwent a screening colonoscopy during 2008–2016 at the Forzani & MacPhail Colon Cancer Screening Centre (*n* = 1083).

Variable	Overall	Cases	Noncases
	1083	215	868
Age (mean (SD))	60 (6.1)	62 (6.4)	60 (6.0)
Diet Parameters (g/d) mean (SD)			
Total Meat Intake	73 (53.2)	81 (57.9)	71 (51.8)
Total Unprocessed Meat Intake	49 (38.2)	52 (39.8)	48 (37.8)
Total Processed Meat Intake	24 (21.4)	29 (26.1)	23 (19.9)
Total Protein Intake	72 (32.5)	73 (37.4)	72 (31.2)
Total Fat Intake	66 (32.6)	67 (35.5)	66 (31.8)
Total Sugar Intake	95 (48.6)	94 (44.0)	95 (50.2)
Total Energy Intake (kcal/d)	1765 (722.0)	1781 (745.1)	1760 (716.6)
Sex *n* (%)			
Female	382 (35.3)	54 (25.1)	328 (37.8)
Male	701 (64.7)	161 (74.9)	540 (62.2)
Body Mass Index (BMI) *n* (%)			
<25	323 (29.8)	49 (22.8)	274 (31.6)
25–30	495 (45.7)	105 (48.8)	390 (44.9)
>30	265 (24.5)	61 (28.4)	204 (23.5)
Alcohol Consumption Status *n* (%)			
Never	64 (5.9)	14 (6.5)	50 (5.8)
Former	55 (5.1)	9 (4.19)	46 (5.3)
Occasional	310 (28.6)	60 (27.9)	250 (28.8)
Daily	654 (60.4)	132 (61.4)	522 (60.1)
Physical Activity *n* (%)			
Low	63 (5.8)	14 (6.5)	49 (5.7)
Mod	295 (27.2)	70 (32.6)	225 (25.9)
High	668 (61.7)	125 (58.1)	543 (62.6)
None	57 (5.3)	6 (2.8)	51 (5.9)
Smoking Status *n* (%)			
Never	515 (47.5)	90 (41.9)	425 (49.0)
Former	461 (42.6)	93 (43.3)	368 (42.4)
Current	107 (9.9)	32 (14.9)	75 (8.6)
Ethnic Group *n* (%)			
White	930 (85.9)	186 (86.5)	744 (85.7)
Other	153 (14.1)	29 (13.5)	124 (14.3)
Diabetes *n* (%)			
No	979 (90.4)	189 (87.9)	790 (91.0)
Yes	104 (9.6)	26 (12.1)	78 (9.0)

SD: standard deviation; kcal/d: kilocalories per day; g/d: grams per day; %: percentage.

**Table 2 cancers-16-00495-t002:** Summary of crude and adjusted logistic model associations between total, processed, and unprocessed red meat intake and the presence of any advanced colorectal adenomas (*n* = 1083).

	OR (95% CI)		
Variable	Total Meat Intake (per 10 g)	Total Processed Meat Intake (per 10 g)	Total Unprocessed Meat Intake (per 10 g)
Model 1	1.03 (1.01–1.06)	1.12 (1.05–1.20)	1.03 (0.99–1.06)
Model 2	1.03 (1.01–1.06)	1.11 (1.04–1.18)	1.02 (0.98–1.06)
Model 3	1.02 (0.99–1.05)	1.09 (1.01–1.17)	1.02 (0.98–1.06)
Model 4	1.05 (1.01–1.10)	1.12 (1.03–1.22)	1.04 (0.98–1.09)

CI: confidence interval; g: grams. Model 1 was unadjusted; Model 2 adjusted for sex (males/females) and age (continuous); Model 3 includes covariates for Model 2 as well as BMI (<25, 25–30, 30 kg/m^2^), smoking status (never, former, current smoker), alcohol consumption (daily, former, occasional, never), ethnicity (white, other), diabetes (yes/no), physical activity (inactive, low, moderate, and high activity); Model 4 includes covariates from model 3 as well as total energy intake (kcal/d), protein and fat intake (g/d).

## Data Availability

The data sets generated during and/or analyzed during the current study are not publicly available. Researchers with approved ethics applications may apply to collaborate and access through the Forzani & MacPhail Colon Cancer Screening Centre.

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
