# Peer review of "The Association between Red Meat Consumption and Advanced Colorectal Adenomas in a Population Undergoing a Screening-Related Colonoscopy in Alberta, Canada"

_cancers, 2024, doi:10.3390/cancers16030495_

Round 1

Reviewer 1 Report

Comments and Suggestions for Authors

The manuscript is well written, however minor changes are required. Adding some statistics regarding CRC and adenomas from Canada would make the introduction stronger. 

In the results section Table 1, correct the decimals and make it consistent throughout the table. In line 156, check the number for the daily energy consumption. 

Checking interaction with smoking, bmi and other potential risk factors will make the manuscript stronger. 

Author Response

The association between red meat consumption and advanced colorectal adenomas in a population undergoing screening-related colonoscopy in Alberta, Canada

Manuscript ID: cancers-2814476

We want to thank the reviewers for the thoughtful comments, which have allowed us to improve our paper. On behalf of my co-authors, we are enthusiastic about the opportunity to revise and resubmit our manuscript “The association between red meat consumption and advanced colorectal adenomas in a population undergoing screening-related colonoscopy in Alberta, Canada.” for consideration in Cancers.

Below we have provided our point-by-point responses to the reviewer comments, as well as a summary of changes/modifications in the revised version of our manuscript.

Reviewer #1

  1. COMMENT: The manuscript is well written, however minor changes are required. Adding some statistics regarding CRC and adenomas from Canada would make the introduction stronger.

RESPONSE: We appreciate the reviewer's suggestion and have taken steps to enhance the introduction. In response to this feedback, we have incorporated statistics from the Canadian Cancer Society regarding CRC incidence and mortality in Canada to provide a stronger context for our study. Additionally, we have included an introductory paragraph that explains why assessing red meat intake is essential as it represents a modifiable risk factor for colorectal adenomas. This preamble provides readers with a clearer understanding of the relevance of our research.

  1. COMMENT: In the results section Table 1, correct the decimals and make it consistent throughout the table. In line 156, check the number for the daily energy consumption.

RESPONSE: We thank the reviewer for their attention to detail. We have addressed this concern by ensuring that decimals in Table 1 are corrected and consistently formatted. Additionally, we have verified and corrected the daily energy consumption figure to 1765 kcal in line 156.

  1. COMMENT: Checking interaction with smoking, BMI, and other potential risk factors will make the manuscript stronger.

RESPONSE: We evaluated the interactions between the red meat variables of interest and smoking and BMI in our linear regression model by creating multiplicative interaction terms.  None of the interactions were significant at the level of p<0.1 and therefore we did not include these in the manuscript.  We did note these additional analyses in the manuscript.

Reviewer 2 Report

Comments and Suggestions for Authors

Cancers

cancers-2814476

The association between red meat consumption and advanced colorectal adenomas in a population undergoing screening-related colonoscopy in Alberta, Canada

Dear Editor,

The paper deals with the investigation of whether the consumption of red meat, processed meat and unprocessed meat contribute to the risk of developing advanced colorectal adenomas. The topic is really good. The manuscript has been generally well designed and written. However, the discussion section should be improved. In addition, do researchers have data after 2016? My comments and questions;

-       Please discuss the differences between processed and unprocessed meat!

-       Line 78: Is there data after 2016?

-       Line 98: Please give the questionnaire about data collection!

-       Line 156: Please check the energy value, 17564 kcal/d? I think it is too much and does not fit the values in Table 1.

-       Table 1: How did the authors determine total protein, fat, sugar, and energy intake, as well physical activity?

-       Lines 243 and 244: Why?

Comments on the Quality of English Language

It is fine.

Author Response

The association between red meat consumption and advanced colorectal adenomas in a population undergoing screening-related colonoscopy in Alberta, Canada

Manuscript ID: cancers-2814476

We want to thank the reviewers for the thoughtful comments, which have allowed us to improve our paper. On behalf of my co-authors, we are enthusiastic about the opportunity to revise and resubmit our manuscript “The association between red meat consumption and advanced colorectal adenomas in a population undergoing screening-related colonoscopy in Alberta, Canada.” for consideration in Cancers.

Below we have provided our point-by-point responses to the reviewer comments, as well as a summary of changes/modifications in the revised version of our manuscript.

Reviewer #2

  1. COMMENT: Please discuss the differences between processed and unprocessed meat!

RESPONSE: We thank the reviewer for their suggestion. We have made considerable improvements to address this request. We have introduced additional content in line 269, which serves to highlight the specific characteristics that differentiate processed and unprocessed red meat. Furthermore, in lines 287-303, we have provided comprehensive details outlining the key differences between these two categories.

.

  1. COMMENT: Line 78: Is there data after 2016?

RESPONSE: We appreciate the reviewer's attention to detail. To address this query, we have corrected the time frame in our manuscript. Participants in our study underwent screening colonoscopy from 2008-2020. We thank the reviewer for highlighting this discrepancy and ensuring the accuracy of our timeline.

  1. COMMENT: Line 98: Please give the questionnaire about data collection!

RESPONSE: We appreciate the reviewer's interest in our data collection process. While the Dietary History Questionnaire (DHQ) comprises 165 questions, it is not feasible to include the entire questionnaire due to space constraints. However, to provide insight into our data collection methodology, we have included a representative sample of four questions from the DHQ in the methods section, specifically in lines 128-133. These questions have been included to exemplify the type of information collected through the DHQ and offer readers a glimpse into our data collection approach. We believe that this sample adequately describes the comprehensive nature of the questionnaire while maintaining the manuscript's readability. We have also cited the DHQ in our manuscript, allowing interested readers to refer to the DHQ directly for a complete overview of the questionnaire and its content.

  1. COMMENT: Line 156: Please check the energy value, 17564 kcal/d? I think it is too much and does not fit the values in Table 1.

RESPONSE: We appreciate the reviewer for bringing this error to our attention. Upon thorough review, we have identified and corrected the energy value to 1765 kcal/d in line 156. This correction ensures consistency with the values presented in Table 1. Thank you for helping us maintain accuracy in our manuscript.

  1. COMMENT: Table 1: How did the authors determine total protein, fat, sugar, and energy intake, as well physical activity?

RESPONSE: We appreciate the reviewer's inquiry regarding our methodology for determining nutrient intake and physical activity. We used the NIH-developed software Diet*Calc allowed us to accurately compute daily intake values, encompassing macronutrients like protein, fat, carbohydrates, and kilocalories. Additionally, we considered specific activity types such as walking, moderate-intensity exercises, and vigorous intensity activities. To gauge overall physical activity levels, we collected separate data on activity frequency (measured in days per week) and duration (measured in time per day) for each specific activity type. This integrated approach ensured a robust evaluation of both nutrient intake and physical activity levels in our study participants.

  1. COMMENT: Lines 243 and 244: Why?

RESPONSE: We appreciate the reviewer's question regarding the associations presented in our study. To provide further clarification, we have revised lines 243 and 244 as follows: "Unlike total and processed red meat, the associations between ACRA and unprocessed meat consumption were not statistically significant, possibly due to our study's limited sample size." We have replaced "likely" with "possibly" to emphasize that the small sample size could be one of the reasons, among others, for the associations not reaching statistical significance. Indeed, the limited sample size may have constrained our ability to detect meaningful differences between groups.

Reviewer 3 Report

Comments and Suggestions for Authors

In this study, the authors aimed to explore the association between different types of red meat and the development of early precancerous colorectal lesions. They found that  eating red and processed meats may increase the risk of advanced colorectal adenomas, while unprocessed meats did not show a meaningful association. The conclusions were supported by some solid data. However, a few issues need to be addressed.

1.       The introduction is not strong enough to attract readers.

2.       It is better to have an abbreviation list.

        3. In this research, the sample population also includes family history of adenomatous polyps or CRC subgroups, did the authors tried to analyze any differences between this subgroup and other subgroups?

        4. What is the proportion of  family history of adenomatous polyps or CRC subgroups in cases?

Comments on the Quality of English Language

English language is good.

Author Response

 The association between red meat consumption and advanced colorectal adenomas in a population undergoing screening-related colonoscopy in Alberta, Canada

Manuscript ID: cancers-2814476

We want to thank the reviewers for the thoughtful comments, which have allowed us to improve our paper. On behalf of my co-authors, we are enthusiastic about the opportunity to revise and resubmit our manuscript “The association between red meat consumption and advanced colorectal adenomas in a population undergoing screening-related colonoscopy in Alberta, Canada.” for consideration in Cancers.

Below we have provided our point-by-point responses to the reviewer comments, as well as a summary of changes/modifications in the revised version of our manuscript.

Reviewer #3

  1. COMMENT: The introduction is not strong enough to attract readers.

RESPONSE: We appreciate the reviewer's suggestion and have taken steps to enhance the introduction. In response to this feedback, we have incorporated statistics from the Canadian Cancer Society regarding CRC incidence and mortality in Canada to provide a stronger context for our study. Additionally, we have included an introductory paragraph that explains why assessing red meat intake is essential as it represents a modifiable risk factor for colorectal adenomas. This preamble provides readers with a clearer understanding of the relevance of our research.

  1. COMMENT: It is better to have an abbreviation list.

RESPONSE: Thank you for your recommendation. We have included an abbreviation list to improve the clarity of our manuscript. Abbreviations include:

CRC - Colorectal Cancer

ACRA - Advanced Colorectal Adenomas

DHQ - Diet History Questionnaire

IPAQ - International Physical Activity Questionnaire

BMI - Body Mass Index

OR - Odds Ratio

CI - Confidence Interval

kcal/d - Kilocalories per day

g/day - Grams per day

CRA - Colorectal Adenomas

NOCs - N-Nitroso Compounds

HCAs - Heterocyclic Amines

PAHs - Polycyclic Aromatic Hydrocarbons

We hope this list enhances the clarity and readability of our manuscript for readers.

  1. COMMENT: In this research, the sample population also includes family history of adenomatous polyps or CRC subgroups, did the authors try to analyze any differences between this subgroup and other subgroups?

RESPONSE: We appreciate the reviewer's inquiry. It's important to clarify that all participants in our study are at an average-risk, meaning they do not have a family history of CRC, specifically, no first-degree relatives with CRC. This information has been added in both line 98 and line 104 of our manuscript to better define our study population. Consequently, we did not analyze differences between subgroups with a family history of adenomatous polyps or CRC and other subgroups since our study specifically focuses on individuals at average risk for CRC. We hope this explanation addresses the reviewer's query and provides clarity regarding our study population.

  1. COMMENT: What is the proportion of family history of adenomatous polyps or CRC subgroups in cases?

RESPONSE: We sincerely thank the reviewer for their question. In our study, all participants are considered to be at average risk for CRC, which means they do not have a family history of adenomatous polyps or CRC, specifically, no first-degree relatives with CRC. As a result, there is no proportion of individuals with a family history of adenomatous polyps or CRC in our cases. Our research specifically focuses on individuals at average risk for CRC.

Round 2

Reviewer 3 Report

Comments and Suggestions for Authors

The authors tried to explore the associations of red meat consumption and the risk of colorectal cancer, which is important in our daily life. They provide solid data to support the conclusions and highlights the importance of differentiating between types of red meat consumption in the context of dietary risks associated with ACRA. This is a meaningful research for people to learn more about CRC and food.